# Content and Speciation of Phosphorus in Lake Kórnickie

Ewelina Janicka [1],*, Jolanta Kanclerz [1] and Katarzyna Wiatrowska [2]

[1] Department of Land Improvement, Environmental Development and Spatial Management, Poznań University of Life Sciences, Piątkowska 94, 60-649 Poznań, Poland

[2] Department of Soil Science, Land Reclamation and Geodesy, Poznań University of Life Sciences, Piątkowska 94, 60-649 Poznań, Poland

* Correspondence: ewelina.janicka@up.poznan.pl

**Abstract:** This paper presents the speciation of phosphorus in bottom sediments and its spatial variability in Lake Kórnickie. This study provides a quantitative determination of the abundance and chemical speciation of P and potential P-release rates from Lake Kórnickie. Phosphorus (P) is an important macronutrient that can limit primary productivity in fresh water ecosystems. The study was conducted during the hydrological years 2016–2018. The speciation analysis was carried out using Visual Minteq software. The predominant form of orthophosphate (V) in the waters of Lake Kórnickie was the $HPO_4^{2-}$ anion, which was related to the reaction of the studied waters. Conditions favoring the precipitation of orthophosphates to hydroxyapatite, aragonite, and calcite prevailed in the lake waters. No cyclic periods of deposition of minerals and release of phosphorus from bottom sediments were observed and, for most of the study period, the lake acted as a "trap" for phosphorus at point JK1. The findings of this study suggest that the internal sedimentary P loading contributes substantial bioavailable P to the water column at one of sampling points (JK2). The accumulation of phosphorus in bottom sediments meant that the lake restoration processes carried out in various lakes may not have had the intended results. At the same time, bottom sediments can be a secondary source of pollution of river waters with this element despite the reduction of inflow from the catchment area of this element.

**Keywords:** lake; geochemistry of phosphorus; speciation of phosphorus



## 1. Introduction

Lake eutrophication is accelerated by nutrient pollution, one of the most widespread water quality problems worldwide [1,2]. Phosphorus (P) along with nitrogen (N) is the key element responsible for the eutrophication process, which accumulates in bottom sediments. For most inland waters, this element plays an important role in regulating productivity and water quality and determines biological productivity [3]. The biogeochemical cycle and availability of phosphorus depend largely on its chemical forms in the aquatic ecosystem. Taking into account this fact, knowledge of the speciation forms of phosphorus in the water body provides an opportunity to understand the role of phosphorus in water quality regulation [4,5]. From the point of view of increasing eutrophication processes, it is important to understand the processes taking place in reservoirs, as knowledge in this area helps guide lake management practices.

Phosphorus found in bottom sediments usually occurs in various forms, the most common being phosphorus bound to iron, calcium, or organic matter [6,7]. The speciation forms of orthophosphate that are most common in bottom sediments are calcite, dolomite, and hydroxyapatite. Fractions associated with calcium are also very common and their distribution largely depends on the substrate on which they occur [8]. The interaction of water and sediment is of special importance, as sediment can also release heavy metals and act as a source of pollution. Bottom sediments accumulating in lakes can act as a sink for pollutants in lakes. These pollutants in the form of phosphorus, nitrogen, or heavy metals

can be assimilated into bottom sediments [9,10]. Sometimes, however, bottom sediments act in the opposite direction, releasing nutrients. Bottom sediments can contribute phosphate loads to water at levels comparable to external sources of the element. In surface waters, phosphorus can be released in the form of dissolved minerals such as hydroxyapatite [8,11–14]. The phenomenon of phosphorus accumulation or release is determined by numerous biological (bacterial activity and mineralization processes), physicochemical (desorption and dissolution), physical (diffusion, sediment mixing, and water temperature), and chemical (dissolved oxygen concentration, redox potential, pH, and iron/phosphorus ratio) processes, which depend, among other things, on the morphometry of the lake, especially the depth [14–17].

Quantifying the speciation of phosphorus in bottom sediments helps to understand its potential mobility and the possibility of long-term phosphorus release. It also helps assess the mobility of specific fractions that are considered inert in the aquatic environment. Examples of such fractions are apatite (HCl-P) and aluminum-bound (NaOH-rP), whose mobility can increase as a result of certain environmental conditions such as high pH, as suggested by numerous studies [18–21]. Other scientists emphasize the importance of the presence of calcite and dolomite (calcareous minerals), the presence of which in the water can affect the phosphorus compound processes taking place. If this is the case, the phosphorus present in the bottom sediments can both adsorb and precipitate calcite, and thus can absorb or release phosphate [22].

This paper attempts to determine the speciation forms of phosphorus present in bottom sediments and to determine their mobility. In addition, the relationship between speciation forms and surface water chemistry was analyzed and the role of phosphorus present in sediments was determined. The subject of the content of biogenic elements in lake waters is a widely reported concept in the global literature, while reports on the content and speciation of phosphorus in lakes are less common. The topic of speciation forms is not a popular topic; the quality of reservoirs most often refers to the topic of quality analysis in terms of physicochemical indicators. Moreover, there are no scientific reports in this field for Lake Kórnickie. In this paper, we (1) study P speciation in bottom sediments and spatial variability in Lake Kórnickie, (2) determine the amount of mobile fractions, (3) present the relationship of P fractions formed in bottom sediments and the content of selected physicochemical indicators determined in the filtrate water, and (4) investigate the role of P release from bottom sediments in the lake and determine how P release from bottom sediments affects the water quality of the river-lake network.

## 2. Materials and Methods

### 2.1. Study Site Description

The study site is Lake Kórnickie, located in the central-western part of Poland, in Wielkopolskie Province, about 30 km south of Poznan (52°14′24′′ N, 17°05′10′′ E) (Figure 1). Lake Kórnickie is a shallow lake in Poland, which, owing to its hydrographic (one of the lakes of a river-lake network) and physiographic (excellent location for tourism) characteristics, is exposed to the process of eutrophication. According to the physico-geographical division of Poland, it is located in the South Baltic Lakelands [23]. The lake is situated in the catchment area of the Głuszynka river and is one of eight lakes of the Kórnicko-Zaniemyska Gutter. The lake has two main tributaries, of which the Głuszynka is the only outflow river that exchanges water in Lake Kórnickie.

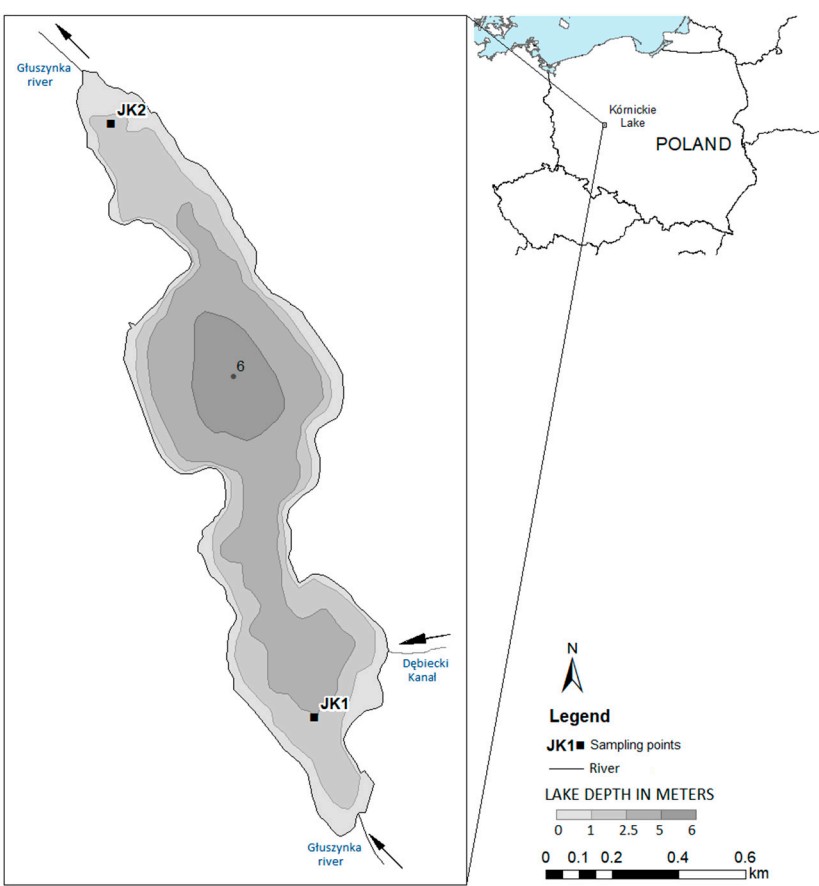

**Figure 1.** Study site location.

The lake has a glacial origin, which is responsible for its ribbon shape. The study area is located within the range of the Weichselian glaciation. The catchment area is covered with luvisols, the largest percentage of which are soils composed of glacial till and loamy sand. Lake Kórnickie, with an area of 85 ha, is classified as a lowland, limestone or mixed, very shallow, and non-stratified lake, and its catchment area is located in a suburban area. The lake plays an important role in fishery and tourism [24].

Since the 1990s, with rapid economic growth, agricultural runoff has seriously contaminated the lake. Since then, severe eutrophication has been a major and consistent environmental problem in Lake Kórnickie and the lake is not being treated, yet. On the other hand, the commune of Kórnik is investing in the revitalization of the area around the lake, with the construction of a promenade and a pier over the lake, among others [25,26].

*2.2. Materials*

The cartographic works involved an analysis of the study area, including the analysis of a structure of the catchment area according to Corine Land Cover 2018 plus an analysis of the terrain relief, a morphometric analysis of Lake Kórnickie. A digital elevation model (DEM) was developed on the basis of LIDAR data in a regular grid with a mesh of 1 m. The possibility of phosphorus accumulation/transport/erosion from the lake bottom sediments was determined, using the methodology of effective fetch calculation (using wind direction data and bathymetric data). All spatial analyses were performed using ArcMap 10.7.1.

As part of the cartographic works, an analysis of meteorological conditions was also conducted by performing thermal and humidity classification based on the data obtained from the Institute of Meteorology and Water Management in Poznań for the Kórnik meteorological station, for the period from 1990 to 2019. Moisture analyses of individual years were performed according to the methodology of Kaczorowska [27], while thermal analyses were performed according to the methodology of Lorenc [28].

### 2.3. Field Works

The samples of water (above the surface of the bottom sediments) for the laboratory analyses were collected from two sampling points (SPs) located on Lake Kórnickie and labelled with an alphanumeric code (JK1–JK2) (Figure 1).

At a frequency of once per month during hydrological years in the period 2016 to 2018, the water samples were collected in 1 L polyethylene bottles. The samples were then transported to the laboratory where analyses were performed over a period of 48 h.

Measurements of the water temperature, oxygen content, pH, and EC were performed in situ using the CC-105 by ELMETRON [29]. Physicochemical elements were compared in accordance with the applicable of European Communities Environmental Objectives (Surface Water) Regulations 2009 [30]. The results were compared with the guidelines for lakes. However, in the absence of guidelines for lakes, the guidelines for rivers were taken into account (Lake Kórnickie is a flow-through lake; the Głuszynka River flows through Lake Kórnickie).

### 2.4. Laboratory Works

The following parameters were determined in the collected samples of water: $N-NO_3$, $N-NO_2$, $N-NH_4$, $H-CO_3^-$, $Cl^-$, $SO_4^{2-}$, P, $P-PO_4$, $BOD_5$, $Na^+$, $K^+$, $Mg^{2+}$, and $Ca^{2+}$. The measurements were performed in accordance with the current standard. The analytical procedure is described in previous papers [31–37]. The analyses were performed in duplicate and the data were presented as averaged values. These physicochemical elements were compared in accordance with the applicable of European Communities Environmental Objectives (Surface Water) Regulations 2009 [30].

### 2.5. Geochemical Model

Speciation forms of phosphorus in the waters of Lake Kórnickie were estimated using Visual MINTEQ ver. 3.1. The Visual MINTEQ program is a tool designed to calculate metal speciation, solubility equilibria, sorption, and so on for natural waters. Visual MINTEQ is based on constant chemical reactions [38]. In water, speciation of elements implies their distribution in the form of free ions or complexes, which can be determined using speciation geochemical models. Measurement data obtained from two SPs located in the analyzed lake were introduced to the geochemical model. The model takes into account parameters such as water temperature, pH, electrolytic conductivity, and dissolved oxygen concentration, as well as $N-NO_3$, $N-NO_2$, $N-NH_4$, $H-CO_3^-$, $Cl^-$, $SO_4^{2-}$, P, $P-PO_4$, $BOD_5$, $Na^+$, $K^+$, $Mg^{2+}$, and $Ca^{2+}$. On the basis of the entered measurement data, the program generates speciation forms of phosphates and information on the exceeded values of the solubility product of individual minerals. Relying on this, the data of mineral precipitation processes (accumulation of P in bottom sediments) or the release of this element from the bottom sediments of the lake were determined.

### 2.6. Statistical Analyses

Clustering time series were performed to visualize the physical and chemical analyses for the lake. Data were normalized to show similarity and differences of 17 parameters over 19 measurement dates (Euclidean measure of similarity/differences between delineated groups) and then, using the clustering algorithm, dendrograms were developed. These analyses were performed in the program R. Furthermore, heatmap analyses were performed to reveal similarities and differences between physicochemical parameters and speciation forms for the two SPs JK1 and JK2 with statistical software. However, principal component analysis (PCA) was performed to identify factors differentiating the discussed speciation forms and Lake Kórnickie in terms of physicochemical water parameters in the program Statistica 13.3 (TIBCO Software Inc., Palo Alto, CA, USA) [39].

## 3. Results

### 3.1. Characteristics of the Catchment Area

Lake Kórnickie is one of the lakes of the Kórnicko-Zaniemyska Gully. The lake has a maximal length of 2.41 km and maximal width of 0.63 km. A recent bathymetric survey indicated an average water depth of 2.6 m (Figure 1) [26]. The catchment area of Lake Kórnickie with Catchment of Dębiecki Canal is 11.194 km$^2$. Choiński et al. [40] estimated a lake volume of 2164.7 × 103 m$^3$ and water residence of 541%. Lake Kórnickie is supplied by one watercourse, Dębiecki Canal, and the only watercourse flowing out of Lake Kórnickie is the Głuszynka river. The land use pattern in the catchment of Lake Kórnickie is about 54% (6.055 km$^2$) of the total area used for agriculture (mostly non-irrigated arable land—211 code), while forests occupy 21.52% (2.409 km$^2$) of the area (mixed forest—code 313) and artificial surfaces occupy 17.15% (1.920 km$^2$) of the area (discontinuous urban fabric—code 112)—own research (Figure 2). Anthropogenic activities on the lake catchment are known to seriously threaten the Lake Kórnickie environment. The anthropogenic impact is majorly related to agriculture (farmland) and the town of Kórnik. The town of Kórnik is located along the eastern shore of the lake, while Kórnik-Bnin is located in the south. Apart from the landscape value of Lake Kórnickie, in the northern part of the lake, there is a sports and recreation center. On the other hand, the western shore of the lake is covered with an experimental forest of the Polish Academy of Sciences with an area of about 220 ha. According to the information provided by the Regional Inspectorate for Environmental Protection (WIOŚ) in Poznań, in the 1970s, sewage from the dairy plant in Śrem—branch in Kórnik and from G.S. Samopomoc Chłopska—that is, from the gas water plant, bakery, butcher's shop, and municipal sewage [41], and until March 1997, sewage from Masarnia in Kórnik was discharged into the lake [42].

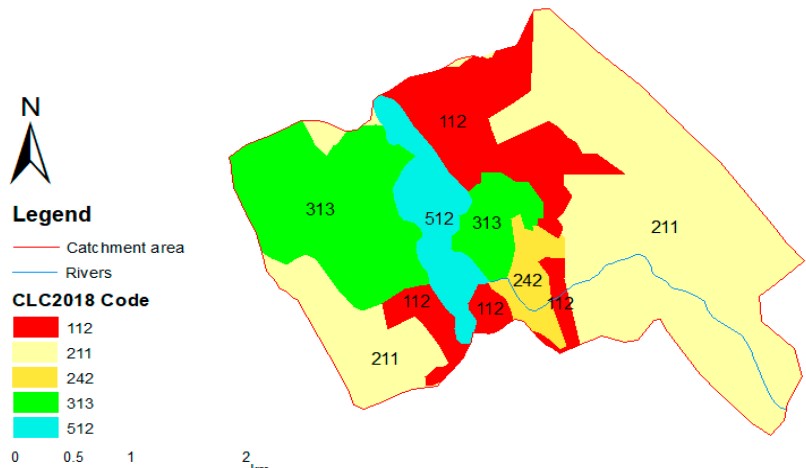

**Figure 2.** Structure of the land use catchment area of Lake Kórnickie (112—discontinuous urban fabric; 211—non-irrigated arable land; 242—complex cultivation patterns; 313—mixed forest; 512—water bodies).

The analyzed catchment area is characterized by the lowest annual precipitation in Poland. The climate of this region is determined by the oceanic influence, with short, mild winters; early springs; and long summers as well as a small annual amplitude of air temperature [43]. The average annual rainfall recorded between 1989 and 2018 was 573 mm and the average annual air temperature during those years was 9.4 °C (48.92 °F)—own research.

The relief of the Lake Kórnickie catchment area is diverse; the variation in ground elevation is 22.69 m (Figure 3). The greatest diversity of the area is observed for the eastern part of the Lake Kórnickie catchment, where there is a watercourse feeding the Dębiecki Canal, which may cause the area to be exposed to surface erosion, and the influx of phosphorus may be the dominant factor.

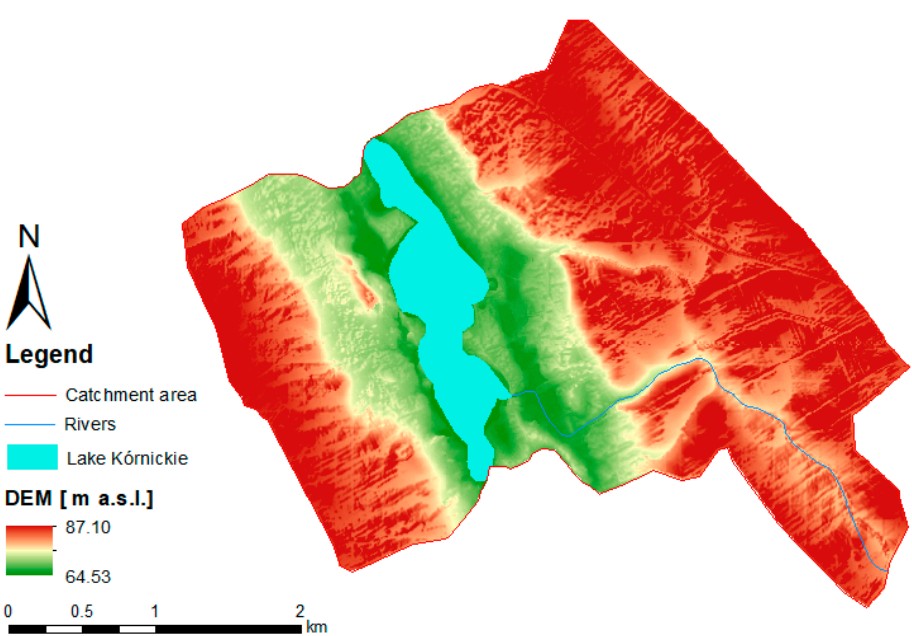

**Figure 3.** Digital elevation model of the catchment area of Lake Kórnickie.

In accordance with the calculation methodology of effective fetch, it was established that, in both SPs, there are conditions favoring the accumulation of phosphorus in the bottom sediments (the effective fetch for JK1 was 175.9 ft and for JK2 was 165.9 ft).

### 3.2. Water Properties

Time series analysis performed for normalized values of examined physicochemical indices for the two SPs JK1 and JK2 showed that the orthophosphate content at both measurement points was the most related to oxygen conditions in the lake (oxygen and $BOD_5$ content). It was observed that, in winter periods (terms 1, 6, 7, 8, 14, and 15), the normalized orthophosphate values were lower than in summer periods and, for winter periods, higher normalized dissolved oxygen and $BOD_5$ values were observed owing to the lower intensity of the photosynthesis process (Figure 4).

The content of orthophosphates at the Lake Kórnickie SP ranged from 0.003 to 0.459 mg $P-PO_4$ at JK1 and from 0.015 to 0.403 mg $P-PO_4$ at JK2. The average content of orthophosphates in the water of Lake Kórnickie at point JK1 was 0.210 mg $P-PO_4$ and at point JK2 was 0.180 mg $P-PO_4$. According to European Communities Environmental Objectives (Surface Waters) Regulations [30], the average orthophosphate content for a good status should not exceed 0.035 mg $P-PO_4$. This means that the average content of orthophosphates in Lake Kórnickie exceeded the guidelines for surface waters (Figure 5). In addition, pH, dissolved oxygen, and $BOD_5$ (these are physicochemical indicators included in the Regulation) were analyzed. For the analyzed samples, the average pH was 8.57 for SP JK1 and 8.42 for SP JK2. This means that the pH of the water met the regulation's requirements for good surface water status. The oxygen content was also up to a good status; the average saturation for both points was within the limits and did not exceed 120%. Unfortunately, in the case of $BOD_5$, it was observed that the average oxygen content was 6.1 $mg \cdot dm^{-3}$ at SP JK1 and 6.09 $mg \cdot dm^{-3}$ at SP JK2. This means that this parameter did not comply with the requirements for good water status.

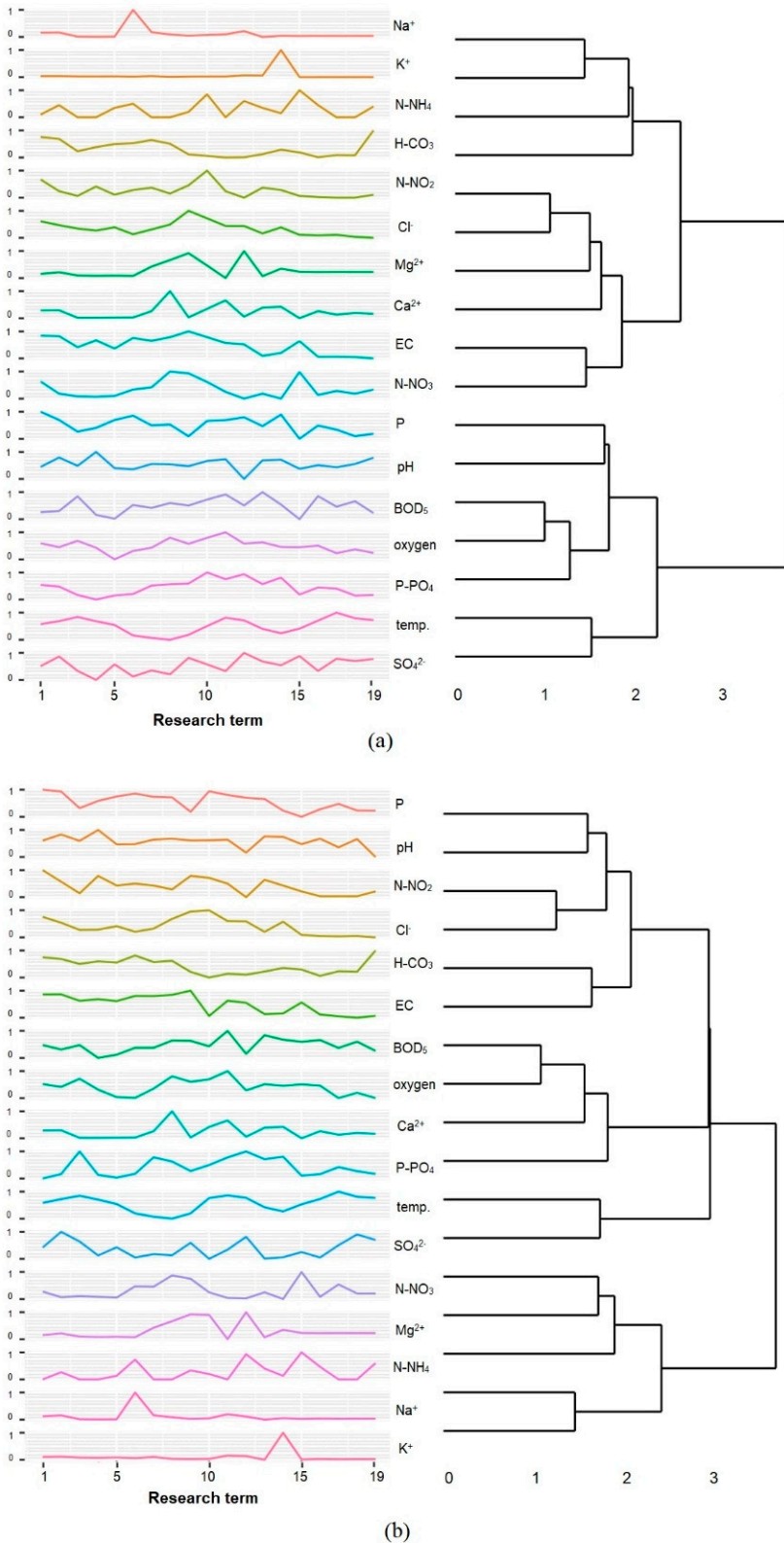

**Figure 4.** Clustering time series for Lake Kórnickie at SPs JK1 (**a**) and JK2 (**b**).

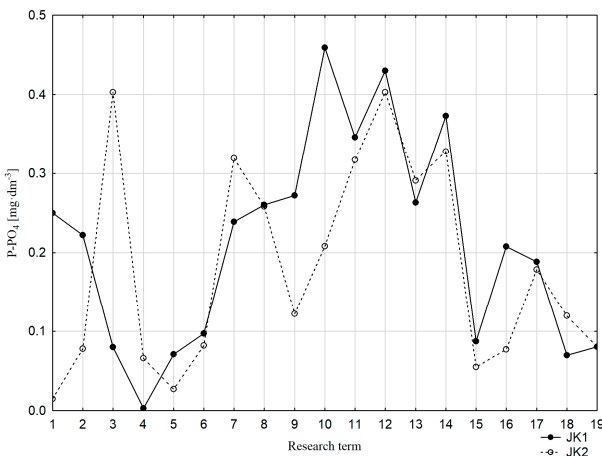

**Figure 5.** Orthophosphate content at sampling points in Lake Kórnickie.

### 3.3. Speciation Forms and Correlation with Surface Water

It was observed that, throughout the study period, phosphates were mainly present in the form of $HPO_4^{2-}$, $H_2PO_4^-$, $CaHPO_4(aq)$, $CaPO^{4-}$, and $MgHPO_4(aq)$, with the $HPO_4^{2-}$ form predominating. The highest value of the $HPO_4^{2-}$ fraction was observed on the sixth measurement date (November 2016) and was 74.059% for JK1 and 74.063% for JK2 (Figure 6). The average content of the $HPO_4^{2-}$ fraction throughout the study period was 52.16% for JK1 and 52.31% for JK2 (Figure 7).

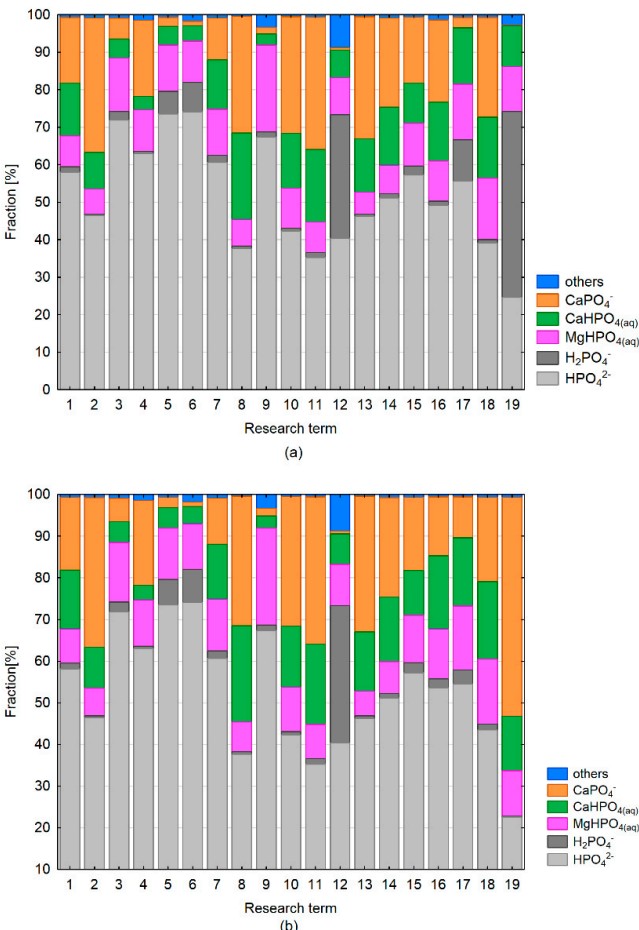

**Figure 6.** Speciation forms of orthophosphates (V) in the waters of Lake Kórnickie at SPs JK1 (**a**) and JK2 (**b**).

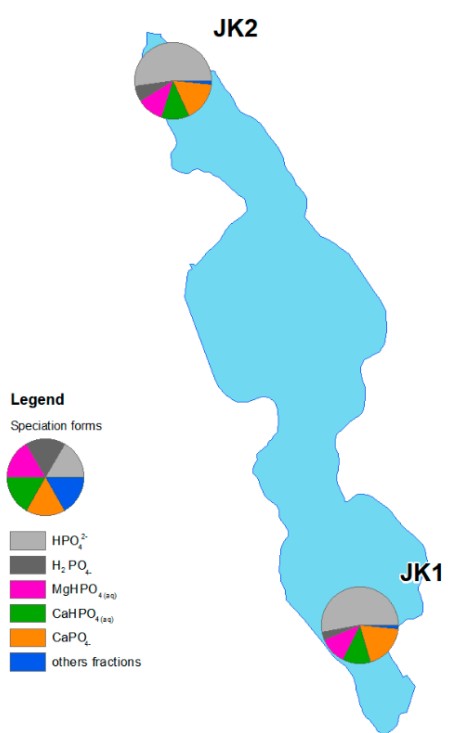

**Figure 7.** Average content of speciation fractions during the study period.

In addition, PCA was used to identify the environmental factors (physicochemical indicators) influencing the content of speciation forms. The aim of PCA was to identify factors that determine the content of speciation forms in bottom sediments. PCA identified two significant factors, factor 1 (first principal components) and factor 2 (second principal components), whose eigenvalues were higher than 1. This analysis for point JK1 explained 79.88% of the variability in the data. The first factor, which accounted for 50.97% of data variability, was mainly related to the fractions of positively correlated indices such as $CaHPO_4(aq)$ and $CaPO^{4-}$. These indicators were negatively correlated with the $HPO_4^{2-}$ form. This indicates that a decrease in the form of $HPO_4^{2-}$ is observed with the growth in the calcium speciation form. The above-mentioned factors were mainly related to indicators such as $Ca^{2+}$, pH, $BOD_5$, and oxygen. The second factor, accounting for 28.91% of the variability in the data, was associated with the form $H_2PO_4^-$ and the other speciation forms. This factor was connected with indicators such as total P, $P-PO_4$, $Mg^{2+}$, $SO_4^{2-}$, and $N-NH_4$. For the second point, JK2, this analysis explained 78.21% of the variability in the data. The first factor, which accounted for 47.31% of data variability, was mainly related to the fractions of positively correlated indices such as $CaHPO_4(aq)$ and $CaPO^{4-}$. These indicators were negatively correlated with the $HPO_4^{2-}$ form (as for point JK1). These factors were mainly related to Ca concentrations. In addition, the second factor, accounting for 30.90% of the variability in the data, was associated with the form $HPO_4^{2-}$. These factors were connected with indicators such as $Mg^{2+}$, $SO_4^{2-}$, and $N-NH_4$ (as for point JK1) and with EC (Figure 8).

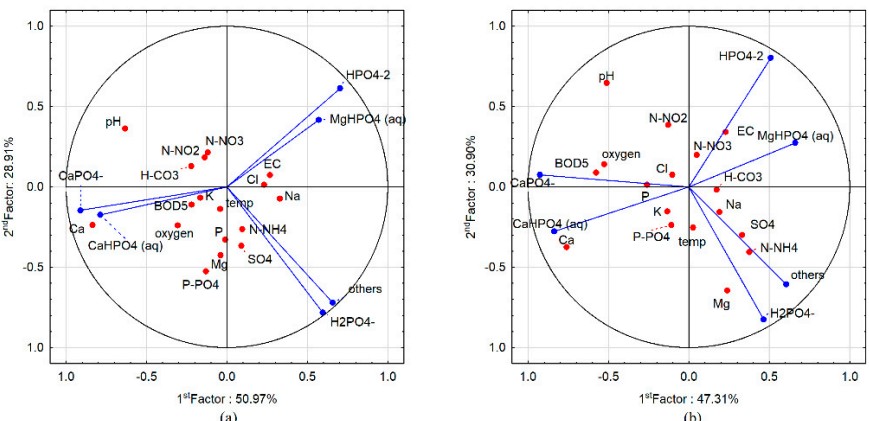

**Figure 8.** Analysis of the main components of the speciation forms of orthophosphates (V) in the waters of Lake Kórnickie in relation to physicochemical parameters in water of Lake Kórnickie; (**a**) for SP JK1 and (**b**) for SP JK2.

In addition, heat maps were generated for the speciation fractions and the indicators extracted in PCA that had the greatest influence on the content of the analyzed fractions (Figure 9).

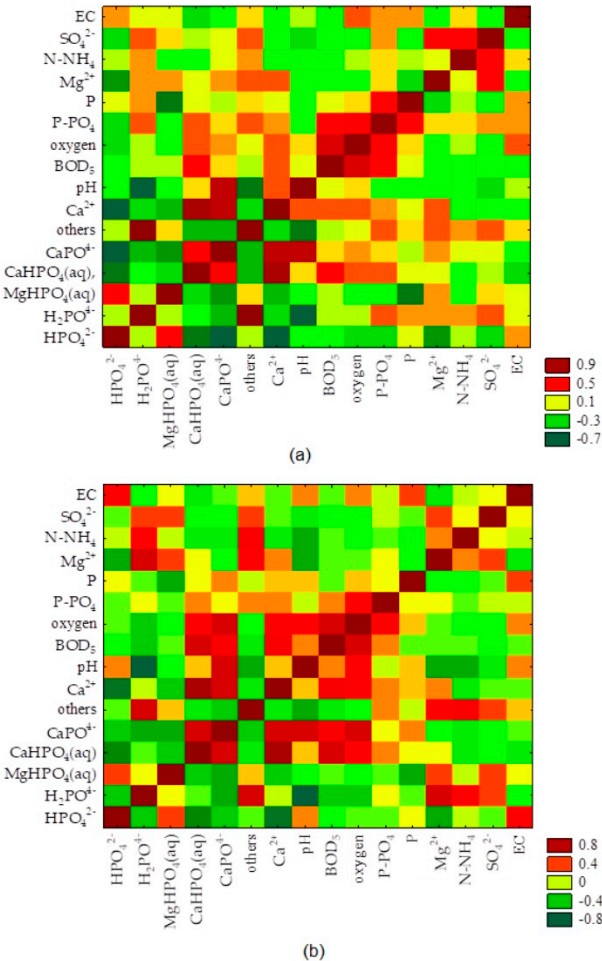

**Figure 9.** Heatmaps for the main components of the speciation forms of orthophosphates (V) in the waters of Lake Kórnickie in relation to physicochemical parameters in water of Lake Kórnickie; (**a**) for SP JK1 and (**b**) for SP JK2.

This analysis confirmed the PCA results indicating the highest significance levels for the selected parameters. A high negative correlation between $HPO_4^{2-}$ and calcium contents (ca $-0.8$) was observed at both JK1 and JK2, which means that high Ca contents resulted in low $HPO_4^{2-}$ contents. On the other hand, the Ca content is strongly influenced by $CaHPO_{4(aq)}$ and $CaPO^{4-}$ contents, where a strong correlation (ca 0.7) was found. Moreover, a strong negative correlation was observed between pH and $H_2PO_4^-$ contents (ca 0.7 at JK1 and 0.8 at JK2).

During the hydrological years 2016–2018, the waters of Lake Kórnickie predominantly experienced conditions conducive to orthophosphate precipitation to hydroxyapatite, aragonite, calcite, $Ca_3(PO_4)_{2(am1)}$, $Ca_3(PO_4)_3 \cdot 3H_2O_{(s)}$, and $CaHPO_{4(s)}$ (Tables S1–S3 for the first three terms). Cyclic periods of precipitation and release of phosphorus from bottom sediments were not observed. For most of the study period, the lake acted as a "trap" for phosphorus at point JK1. In contrast, at SP JK2, there was no precipitation of hydroxyapatite and low concentrations of orthophosphate (V) led to the release of this element from bottom sediments.

## 4. Discussion

In Lake Kórnickie, trends were observed between speciation forms and water chemistry. However, the results obtained indicate the heterogeneous structure of this reservoir. The content of orthophosphates in the water of Lake Kórnickie varied from 0.003 to 0.459 mg P-PO$_4$. According to European Communities Environmental Objectives (Surface Water) Regulations 2009 [30], the average orthophosphate content for good status should not exceed 0.035 mg P-PO$_4$. Unfortunately, 92% of the water samples reached a higher content than the guidelines. The content of orthophosphates is very important from the point of view of speciation forms; it determines the processes of retaining or releasing phosphorus from bottom sediments. During periods of lower orthophosphate content, e.g., on the fourth measurement date, in SP JK1 (did not exceed the regulation for good status) and at a very high pH of 9.68, phosphorus release from bottom sediments was observed. At SP JK2 measurement point, slightly lower orthophosphates contents were observed, which also contributed to the release of phosphorus from bottom sediments into the water column. This was confirmed by the analyses performed in the Visual MINTEQ program, where the release of speciation forms, mainly in the form of hydroxyapatite, was observed. The study proved that the orthophosphate content and pH determined the release or precipitation of phosphorus. This confirms the studies of Moore et al. [15] and Palmer-Felgate et al. [16] or Potasznik and Szymczyk [44], who believe that lake water chemistry influences bottom sediment processes. According to Kanclerz et al. [45], it is the low content of orthophosphate in the water and the lower pH of the water that accounts for the conditions that favor the processes of phosphorus release from bottom sediments. In addition, Zhang et al. [46] in a study of bottom sediments of the Xiaofu river found that, under acidic conditions, it is more likely that Cd, Ni, or Cu will be released into the water.

The speciation analysis conducted for Lake Kórnickie showed that phosphates were most common in the form of $HPO_4^{2-}$, $H_2PO_4^-$, $CaHPO_{4(aq)}$, $CaPO^{4-}$, and $MgHPO_{4(aq)}$, with the predominance of the $HPO_4^{2-}$ form. The proportion of the $H_2PO_4^-$ form was determined by the pH, which was confirmed by PCAs and heat maps. These insights are also confirmed by the study of Spivakov et al. [47], who found that these strong pH increases have been shown to prompt phosphorus release as well. The most frequent soluble forms of phosphorus are orthophosphate under the pH conditions normally (pH $6 \pm 9$) encountered in natural waters and organic phosphorus compounds. Orthophosphates are readily available for assimilation by organisms; they may, however, be removed from the dissolved phase by chemical precipitation with $Al^{3+}$, $Fe^{3+}$, and $Ca^{2+}$. Particulate and dissolved organic phosphorus forms undergo mineralization and the phosphorus is transferred into the soluble orthophosphate pool. Iron(III) phosphates or Fe(III) complexes, which absorb phosphorus, most obviously play an important role in the cycling of phosphorus in the environment.

The results of this paper show that phosphates from bottom sediments occurred during periods of a lower content of orthophosphates in water and this led to the dissolution of hydroxyapatite and other minerals of this element ($Ca_3(PO_4)_{2(beta)}$, $Ca_3(PO_4)_{2(am2)}$, and $Ca_3(PO_4)_{2(am1)}$, $Ca_3(PO_4)_3 \cdot 3H_2O_{(s)}$, and $CaHPO_{4(s)}$. The lower content of phosphates in the waters could have been caused by the intensive photosynthesis process and the uptake of nutrients by primary producers.

On the other hand, phosphorus release was observed for the entire study period at point JK2, which was associated with a lower orthophosphate content and pH. Phosphorus release in the form of dissolved minerals occurred mainly as hydromagnesite, anhydrite, and artinite. $KCl_{(s)}$ release was observed on some measurement dates, which is an unusual phenomenon for a temperate climate zone. This process was probably related to the delivery of a large load of potassium ions and chloride anions to the waters, which had favorable conditions for precipitation under low flow conditions. This is confirmed by studies by Golterman [48] and Serrano et al. [49], who stated that apatite dissolution may be an alternative phosphorus-secreting process. In the sediments of eutrophic hard water lakes, apatite formation occurs, which can be caused by direct precipitation, co-precipitation with $CaCO_3$, or adsorption on $CaCO_3$ particles. When the sediments become anaerobic, there is an automatic lowering of pH as a result of $CO_2$ production. This lowering causes apatite to dissolve, as pH strongly controls its solubility. Some of the secreted phosphate will be readsorbed onto $Fe(OOH)$, which is also present in these sediments. Phosphorus release is thus a function of accumulated phosphorus loading and calcium concentration.

Lake Kórnickie is a shallow lake, on a calcareous substrate, which is a reason for the large number of calcareous forms (abiotic type L-CB1; lowland, shallow, calcareous), which, consequently, is the cause of leaching of minerals into the water depths. The influence of substrate on the presence of minerals in water and bottom sediments is emphasized by Spivakov et al. [47]. The authors emphasize that a major burden on reservoirs is the weathering of rocks, whereby minerals in the form of apatite and silicate minerals are released into the water. Rock weathering also produces as reaction products a number of clay minerals with large adsorptive capacities, and orthophosphates as well as some organic phosphorus compounds might be sorbed on the surface of such particles and reach the lake bottom. There are speciation forms associated with calcium, $CaHPO_{4(aq)}$, and $CaPO^{4-}$ in the waters of Lake Kórnickie, which supports this theory. The lake also witnessed precipitation of calcite and dolomite on most measurement dates at both measurement points. According to Wang et al. [22], the presence of calcite and dolomite (calcareous minerals) in over-dense waters can affect the processes involving phosphorus compounds. If this is the case, the phosphorus present in the bottom sediments can both adsorb and precipitate calcite, and thus can absorb or release phosphate. This is confirmed by the present analysis, which showed phosphorus release at SP JK2 and absorption at SP JK1. The behavior of this lake shows that, although the lake in the upper part was a trap for nutrients, as a result of changes in the physicochemical conditions of the water, in the part of the lake located at the outflow, there was a renewed, internal process, enriching the waters. Consequently, the lake was contributing to the deterioration of the water quality of the river flowing through the lake. Phosphorus accumulation in bottom sediments can cause the lake reclamation projects carried out on various lakes to fail to achieve the intended results. At the same time, bottom sediments can be a secondary source of pollution of river waters with this element, despite the reduction in inflow from the catchment area of this element.

## 5. Conclusions

Eutrophication of lake waters depends on many factors. It is a result of the catchment use structure, seasonality of anthropopressure factors, activity of living organisms, and geochemical processes, among others.

The statistical analyses performed in the study showed the relationship of the speciation analysis carried out with the physicochemical analysis of the waters. The main factors influencing the speciation analysis in the case of Lake Kórnickie were water reaction and

orthophosphate content, and the dominant form of orthophosphate (V) in the waters of Lake Kórnickie was the $HPO_4^{2-}$ anion.

In the waters of Lake Kórnickie, there were conditions conducive to the precipitation of orthophosphates in the form of hydroxyapatite, aragonite, calcite, and others. In the analyzed lake, cyclic periods of deposition of minerals and release of phosphorus from bottom sediments were not observed. In the analyzed lake, phosphorus release from bottom sediments was observed at one point (JK2). The analysis of this lake allows us to conclude that, although the lake in the upper part was a "trap" for nutrients, as a result of a change in the physicochemical conditions of the water, in the part near the mouth, there was an internal process causing enrichment of the waters.

The research carried out in this work represents a small part of the possibilities offered by speciation analysis in the context of surface water quality. An analysis of the literature indicates that this type of research is not commonly performed and has never been performed for Lake Kórnickie. Thus, the studies performed fill the gap in this topic in a small way, and additionally leave open the possibility of further studies, both for the other lakes of the Kórnicko-Zaniemyska Gully and other lakes.

**Supplementary Materials:** The following are available online at https://www.mdpi.com/article/10.3390/w14203234/s1. Table S1. Precipitation to sediment or release of orthophosphates (V) from sediment within 1 measuring period. Table S2. Precipitation to sediment or release of orthophosphates (V) from sediment within 2 measuring period. Table S3. Precipitation to sediment or release of orthophosphates (V) from sediment within 3 measuring period.

**Author Contributions:** Conceptualization, E.J., J.K. and K.W.; methodology, E.J., J.K. and K.W.; software, E.J.; validation, E.J.; formal analysis, E.J. and K.W.; investigation, E.J.; resources, E.J., J.K. and K.W.; data curation, E.J.; writing—original draft preparation, E.J. and J.K.; writing—review and editing, E.J.; visualization, E.J.; supervision, E.J. and J.K.; project administration, E.J.; funding acquisition, J.K. All authors have read and agreed to the published version of the manuscript.

**Funding:** The publication was co-financed/financed within the framework of Ministry of Science and Higher Education programme as "Regional Initiative Excellence" in years 2019–2022, Project No. 005/RID/2018/19.

**Data Availability Statement:** The datasets used and analyzed during the current study are available from the corresponding authors upon reasonable request.

**Conflicts of Interest:** The authors declare no conflict of interest. The funders had no role in the design of the study; in the collection, analyses, or interpretation of data; in the writing of the manuscript; or in the decision to publish the results.

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
