# Peer review of "Content and Speciation of Phosphorus in Lake Kórnickie"

_water, doi:10.3390/w14203234_

Round 1

Reviewer 1 Report

The work is very interesting because it describes the still little known problem of internal water supply with phosphorus accumulated in bottom sediments. Although the research is local in nature, it is a case study, but it is worth publishing. I suggest that the introduction discusses the phosphorus fractions that occur in bottom sediments in more detail. I found the work interesting and should be published with minor corrections to the introduction.

Author Response

Thank you very much for your attention, a note was added to the introduction about the most common phosphorus fractions in bottom sediments.

Reviewer 2 Report

The manuscript "Content and speciation of phosphorus in Lake Kórnickie" is interesting work and well presented. However, prior to acceptance some improvement is recommended 

1. Please carefully check grammatical and typo mistakes found in article. Overall manuscript is well written.

2.Abstract and conclusion section could be further improved. 

3. Fig. 9 needs to be improved. 

Author Response

  1. Please carefully check grammatical and typo mistakes found in article. Overall manuscript is well written.

Answer: The work has been subject to grammar correction by Native Speaker

  1. Abstract and conclusion section could be further improved. 

Answer: Abstract and the summary have been improved

  1. 9 needs to be improved. 

Answer: Figure 9 has been corrected

Reviewer 3 Report

The paper entitled Content and speciation of phosphorus in Lake Kórnickie” generally meets scientific requirements, however it  should be improved in some aspects.

First of all, authors should clearly define objective of the study.

Considering previous studies, authors should mention some novel investigations of this topic, both worldwide and Lake Kórnickie”.  

Structure of the paper should be better organized. Selected part from the Results section should be deleted, while the part from Discussion should be moved in Introduction (see comments in manuscript body). Some terms (environmental factors, factors) are not clear and should be explained. Authors provide numbers as results without explanation. It is necessary to compare obtained results with the standards and discuss them in this manner. Authors should emphasize novelty of their research. Conclusions should be expanded with the recommendations for further investigations.

Some references do not support the text and should be deleted and/or replaced with appropriate ones.

Other comments are given in the manuscript body.

Author Response

The paper entitled “Content and speciation of phosphorus in Lake Kórnickie” generally meets scientific requirements, however it  should be improved in some aspects.

  1. First of all, authors should clearly define objective of the study.

Answer: The purpose of the work was defined at the end of the introduction

  1. Considering previous studies, authors should mention some novel investigations of this topic, both worldwide and Lake Kórnickie”.  

Answer: In introdaction autors get information for this aspect. The subject of the content of biogenic elements in lake waters is a widely reported concept in the world literature, while reports on the content and speciation of phosphorus in lakes are less common. The topic of forums speciation is not a popular topic, the quality of reservoirs most often refers to the topic of quality analysis in terms of physicochemical indicators. Moreover, there are no scientific reports in this field for Lake Kórnickie.

  1. Structure of the paper should be better organized. Selected part from the Results section should be deleted, while the part from Discussion should be moved in Introduction (see comments in manuscript body). Some terms (environmental factors, factors) are not clear and should be explained. Authors provide numbers as results without explanation. It is necessary to compare obtained results with the standards and discuss them in this manner. Authors should emphasize novelty of their research. Conclusions should be expanded with the recommendations for further investigations.

Answer: Thank you very much for your attention, as suggested by the reviewer, some of the results were transferred to the discussion, in fact this fragment referred to the discussion and was mistakenly presented in the results. With regard to the comment on the terms appearing in the work (environmental factors, factors), the authors meant: in the case of environmental factors - physicochemical indicators analyzed during the research, this term was explained in the paper. On the other hand, in the case of "factors" appearing in the discussion of PCA, factor 1 and factor 2 are the terms the authors meant the first and second principal components. Referring to the remark about the lack of comparison of the values with the applicable standards - the paper presents the values of orthophosphates, while the regulation does not include the classification for the indicator. In the case of the Lake Kórnickie, in accordance with the regulation, the lake was classified as shallow lake, on a calcareous substrate, which is a reason for the large number of calcareous forms (abiotic type L-CB1- lowland, shallow, calcareous). Currently, in the case of lakes in this group, the regulation only concerns the visibility of the Secchi disc, conductivity, total nitrogen content and total phosphorus content.

  1. Some references do not support the text and should be deleted and/or replaced with appropriate ones.

Answer: In the absence of references, they were added in the test, while inappropriate references were removed. In addition, the list of references was improved.

  1. Other comments are given in the manuscript body.

Answer: Other comments were included in the article. Thank you for numerous comments in the text, the authors have adapted to the comments contained in the text.

Round 2

Reviewer 3 Report

Authors made some improvements, but not significantly; manuscript still need further revision. The main problem is lack of water quality standards, especially for orthophosphate. Authors mentioned that these standards do not exist, but they should use international standards for comparison in the case of lack of national legislation. Discussion the results without those standards has no sense, because it is not possible to determine what is low or high concentration. Please find these standads at the following link: https://epawebapp.epa.ie/licences/lic_eDMS/090151b2804d65aa.pdf

Some references do not support the text and should be deleted and/or replaced with appropriate ones.

Other comments are given in the manuscript body.

Author Response

At the begining, I would like to thank you sincerely for the reviewer's comments, which were valuable during the preparation of this manuscript.

1. The authors have incorporated the reviewer's comments, which were included in the file attached by the reviewer. 

2. In addition, as recommended by the reviewer, the authors have analyzed the physicochemical parameters in accordance with international standards. I would like to thank you very much for the link to the international standards, it allowed me to describe the most important aspect according to the European Communities Environmental Objectives (Surface Water) Regulations 2009. 
According to this Regulations, the average orthophosphate content for good status should not exceed 0.035 mg P-PO4. The average content of orthophosphates in the water of Lake Kórnickie at point JK1 was 0.210 mg P-PO4 and at point JK2 was 0.180 mg P-PO4. This means that the average content of orthophosphates in Lake Kórnickie exceeded the guidelines for surface waters. Unfortunately, 92% of the water samples reached a higher content than the guidelines. The content of orthophosphates is very important from the point of view of speciation forms, it determines the processes of retaining or releasing phosphorus from bottom sediments. In addition, pH, dissolved oxygen and BOD5 (these are physicochemical indicators in-cluded in the Regulation) were analyzed.

3. The cited sources have been corrected as recommended.

4. Corrected errors in references.